Endosymbiont interference and microbial diversity of the Pacific coast tick, Dermacentor occidentalis, in San Diego County, California

Gurfield Nikos nikos.gurfield@sdcounty.ca.gov 1
Grewal Saran 1
Cua Lynnie S. 1
Torres Pedro J. 2
Kelley Scott T. 2
1 Department of Environmental Health-Vector Control Program, County of San Diego , San Diego , CA , United States of America
2 Department of Biology, San Diego State University , San Diego , CA , United States of America
Newton Irene
Electronic publication date: 2017 Apr 13
Publication date: 2017
Volume: 5
Electronic Location ID: e3202
Received 2016 Jul 2; Accepted 2017 Mar 19
Copyright: ©2017 Gurfield et al.
Copyright year: 2017
Copyright holder: Gurfield et al.
License: This is an open access article distributed under the terms of the Creative Commons Attribution License, which permits unrestricted use, distribution, reproduction and adaptation in any medium and for any purpose provided that it is properly attributed. For attribution, the original author(s), title, publication source (PeerJ) and either DOI or URL of the article must be cited.
License URL: https://creativecommons.org/licenses/by/4.0/

Keywords: Ticks, Endosymbiont, Interference, Microbiome

Funding: The authors received no funding for this work.

==============================
The Pacific coast tick, Dermacentor occidentalis Marx, is found throughout California and can harbor agents that cause human diseases such as anaplasmosis, ehrlichiosis, tularemia, Rocky Mountain spotted fever and rickettsiosis 364D. Previous studies have demonstrated that nonpathogenic endosymbiotic bacteria can interfere with Rickettsia co-infections in other tick species. We hypothesized that within D. occidentalis ticks, interference may exist between different nonpathogenic endosymbiotic or nonendosymbiotic bacteria and Spotted Fever group Rickettsia (SFGR). Using PCR amplification and sequencing of the rompA gene and intergenic region we identified a cohort of SFGR-infected and non-infected D. occidentalis ticks collected from San Diego County. We then amplified a partial segment of the 16S rRNA gene and used next-generation sequencing to elucidate the microbiomes and levels of co-infection in the ticks. The SFGR R. philipii str. 364D and R. rhipicephali were detected in 2.3% and 8.2% of the ticks, respectively, via rompA sequencing. Interestingly, next generation sequencing revealed an inverse relationship between the number of Francisella-like endosymbiont (FLE) 16S rRNA sequences and Rickettsia 16S rRNA sequences within individual ticks that is consistent with partial interference between FLE and SFGR infecting ticks. After excluding the Rickettsia and FLE endosymbionts from the analysis, there was a small but significant difference in microbial community diversity and a pattern of geographic isolation by distance between collection locales. In addition, male ticks had a greater diversity of bacteria than female ticks and ticks that weren’t infected with SFGR had similar microbiomes to canine skin microbiomes. Although experimental studies are required for confirmation, our findings are consistent with the hypothesis that FLEs and, to a lesser extent, other bacteria, interfere with the ability of D. occidentalis to be infected with certain SFGR. The results also raise interesting possibilities about the effects of putative vertebrate hosts on the tick microbiome.

Introduction

The Pacific Coast tick, Dermacentor occidentalis Marx (henceforth D. occidentalis) is the most widely distributed tick in California and is found in chaparral and shrubland areas from northern Baja California to California and Oregon (Furman & Loomis, 1984). D. occidentalis is a three-host, hard-shell tick that feeds on a variety of vertebrates, such as rodents, rabbits, cattle, deer, horses and humans. Surveys of this tick have shown its ability to vector human pathogens such as Francisella tularensis (tularemia), Coxiella burnetii (Q fever), Anaplasma phagocytophilum (human granulocytic anaplasmosis), Ehrlichia chaffeensis (human monocytic ehrlichiosis), Rickettsia rickettsii (Rocky Mountain spotted fever, RMSF) and Rickettsia philipii 364D (hereafter R. philipii) as well as the non-pathogenic spotted fever group Rickettsia, R. rhipicephali (Parker, Brooks & Marsh, 1929; Cox, 1940; Lane et al., 1981; Holden et al., 2003; Wikswo et al., 2008; Shapiro et al., 2010). Rickettsia philipii, was originally described as an unclassified Rickettsia found by Bell in D. occidentalis from California (Philip et al., 1978). It is closely related to Rickettsia rickettsii but can be serologically and genetically distinguished (Philip, Lane & Casper, 1981; Karpathy, Dasch & Eremeeva, 2007). Although discovered in 1966, and long suspected of being able to cause disease, it was only recently confirmed to be associated with eschars and lymphadenopathy in people at the site of a tick bite (Lane et al., 1981; Shapiro et al., 2010; Johnston et al., 2013).

Francisella-like endosymbiotic bacteria (FLEs) have also been detected in Dermacentor occidentalis as well as other tick species (Burgdorfer, Brinton & Hughes, 1973; Noda, Munderloh & Kurtti, 1997; Scoles, 2004; Kugeler et al., 2005). FLEs share 16S rRNA gene homology with Francisella spp., are vertically transmitted, have been observed within tick ovaries and Malpighian tubules, and vary by tick species (Rounds et al., 2012). Although Burgdorfer et al. demonstrated pathogenicity of a Francisella endosymbiont derived from Dermacentor andersoni Stiles ticks (previously categorized as Wolbachia persica, Forsman, Sandström & Sjöstedt, 1994) to guinea pigs and hamsters via injection, most FLEs are not transmitted by tick bites and are considered non-pathogenic (Burgdorfer, Brinton & Hughes, 1973; Niebylski et al., 1997).

Interestingly, the inability of different endosymbiotic Rickettsia species to co-infect the same organ in the same tick, called “interference,” has been demonstrated, although the exact mechanisms are unknown. Early studies seeking to understand the epidemiology of RMSF in the Bitterroot Valley in Montana demonstrated that the non-pathogenic tick endosymbiont Rickettsia peacockii (found on the east side of the valley and originally called the East side agent) colonized the ovaries of D. andersoni ticks and excluded pathogenic Rickettsia rickettsii (more prevalent on the west side of the valley) from infecting the ovaries and being transmitted to eggs (Burgdorfer, Hayes & Mavros, 1981). Similarly, studies of Dermacentor variabilis (Say) infected with R. montanensis or R. rhipicephali demonstrated resistance to transovarial transmission of the reciprocal Rickettsia in challenge experiments (Macaluso et al., 2002). Negative influences between co-infecting species of Rickettsia and other symbionts has been suggested to occur in other vectors such as fleas (Azad & Beard, 1998; Jones et al., 2012). Interference has been postulated to have significant effects in altering the distribution of Rickettsia pathogens in the environment and, consequently, the presence of human disease (Burgdorfer, Hayes & Mavros, 1981).

The use of next generation sequencing has allowed deeper exploration into endosymbionts and complex bacterial communities that colonize different tick species (Nakao et al., 2013), their organs (Budachetri et al., 2014; Qiu et al., 2014), different life stages (Carpi et al., 2011) and different states of nutrition (Menchaca et al., 2013; Zhang et al., 2014). Attention to the microbiome of ticks has been driven, in part, by the fact that ticks can transmit the broadest range of diseases of any arthropod and the recognition that tick co-infections can have dramatic consequences on both the tick host and human patient (Clay & Fuqua, 2010). Microbiome studies using next generation sequencing techniques have demonstrated that each species of tick harbors its own unique bacterial community often dominated by Proteobacteria and one or two endosymbionts (Clay & Fuqua, 2010; Ponnusamy et al., 2014; Hawlena et al., 2012; Van Treuren et al., 2015; Narasimhan & Fikrig, 2015). Given these findings, we hypothesized that next generation sequence analysis of Dermacentor occidentalis ticks microbiomes would reveal patterns of interference or exclusion among pathogenic or non-pathogenic bacteria. We also hypothesized that differences among tick microbiomes would be associated with different geographic locations, and that possible reservoirs of tick pathogens could be found by analyzing ticks for the host origin of prior blood meals or by comparing the tick microbiomes to the skin microbiomes of potential host species. To address these hypotheses, we used culture-independent PCR amplification of the 16S rRNA gene and next-generation sequencing (NGS) to determine whether the microbiomes of SFGR-infected ticks differed from non-SFGR-infected ticks, and if this microbial diversity was consistent with a hypothesis of interference. Our results revealed patterns consistent with partial exclusion between SFGR and FLEs and an association of non-endosymbiotic bacteria with geographic locale. Furthermore, the historical blood meal hosts of the ticks were implicated by the composition of bacterial communities within the ticks and were correlated with SFGR infection. While the precise mechanism of the bacterial interactions (i.e., direct or indirect) need elucidation, our results suggest that carriage of certain pathogenic SFGR in ticks could be modulated by other non-rickettsial endosymbionts, providing a potential non-chemical alternative to SFGR control.

Materials & Methods

Sample collection

Adult ticks were collected from February to May 2014 from 4 different areas of San Diego County: Escondido Creek, Los Peñasquitos Canyon, Lopez Canyon and Mission Trails Regional Park by dragging a 1 m2 piece of canvas over grass and chaparral and then capturing the ticks with forceps and placing them in individual sterile microfuge tubes. The ticks were transported live back to the Vector Disease and Diagnostic Laboratory at the San Diego County Operations Center where, by visual examination, their species and sex were determined and cataloged before freezing them at −80 °C.

DNA extraction, PCR amplification and next generation sequencing

Ticks were processed individually throughout all procedures. The ticks were thawed and washed sequentially in 3% hydrogen peroxide, 100% isopropanol, and sterile distilled water for 1 min in each solution. The final distilled water wash was aspirated from the ticks and then the ticks were sectioned sagittally at midline with a sterile scalpel. Half of the tick was saved at −80 °C; the other half was used for DNA extraction. Briefly, 180 μl of ATL buffer (Qiagen, Valencia, CA, USA) and 20 μl of proteinase K were added to each tick and the ticks lysed overnight at 37 °C in an Eppendorf Thermomixer (Hauppauge, NY) with agitation at 1,400 rpm for 15 s every 15 min, before centrifuging the lysate for 3 min at 18,400× g. The supernatant was transferred into a sterile microfuge tube and DNA extracted using a Qiagen DNeasy Blood and Tissue kit in a Qiacube using the DNeasy Blood and Tissue protocol for Tissue and Rodent Tails (Qiagen, Valencia, CA, USA). Negative extraction controls consisted of sterile water processed via the same washing, chopping and extraction procedure used on the ticks.

The ticks were screened for spotted fever group rickettsia using a Power SYBR Green real-time PCR Mastermix kit (Life Technologies, Carlsbad, CA, USA) and primers for the romp A gene (Eremeeva et al., 2003). Reactions were carried out in a total volume of 20 μL composed of 10 µL Power SYBR Green Mastermix, 0.125 μL each of primers RR190.547F (20 μM) and RR190.701R (20 μM), 7.75 μL of nuclease-free water, and 2 μL of template DNA (Eremeeva et al., 2003; Wikswo et al., 2008). Real-time PCR cycling conditions were: 3 min at 95 °C; 40 cycles of: 20 s at 95 °C, 30 s at 57 °C, 30 s at 65 °C; a holding cycle of 5 min at 72 °C; and a continuous cycle of: 15 s at 95 °C, 1 min at 55 °C, 30 s at 95 °C, 10 s at 55 °C; and a final holding temperature of 4 °C.

DNA from ticks that screened positive for SFGR were subjected to semi-nested PCR amplification of rompA using primers Rr190-70, Rr190-701, and Rr190-602 and the intergenic region (IGR) using primary and nested primers RR0155-rpm B (Eremeeva et al., 2006; Shapiro et al., 2010; Wikswo et al., 2008). Briefly, 20 μL of 2X Taq Master Mix (Qiagen, Valencia, CA), 2 μL of forward primer Rr190-70 (20 mM), 2 μL of reverse primer Rr190-701/Rr190-602 (20 mM), 14 μL of nuclease-free H2O, and 2 μL of DNA was amplified using PCR cycling conditions of 95 °C for 3 min followed by 35 cycles of 95 °C for 20 s, 57 °C for 30 s, and 68 °C for 2 min and then 72 °C for 5 min before holding the products at 4 °C. For the IGR PCR amplification, 20 μL of 2X Taq Master Mix (Qiagen, Valencia, CA, USA), 1 μL of forward primer RR 0155 PF (20 mM), 1 μL of reverse primer 0155 PR (20 mM), 16 μL of nuclease-free H2O, and 2 μL of DNA was amplified using PCR cycling conditions of 95 °C for 5 min followed by 35 cycles of 95 °C for 30 s, 50 °C for 30 s, and 68 °C for 1 min and then 72 °C for 7 min before holding the products at 4 °C.

Amplification products were visualized in a 1% agarose gel stained with ethidium bromide on a UV illuminator and subsequently purified using the PureLink PCR Purification Kit, following the manufacturer’s protocol (Life Technologies, Carlsbad, CA, USA). Products were sequenced using the BigDye Terminator v3.1 Cycle Sequencing Kit and purified using the BigDye XTerminator Purification Kit following the manufacturer’s protocols on an AB 3500xL Genetic Analyzer (Applied Biosystems, Grand Island, NY, USA). Due to highly conserved 16S rRNA gene sequences between Francisellaceae, DNA extracts of the ticks were also tested specifically for the presence of Francisella tularensis using a multi-target real-time PCR test employing primers ISFtu 2, iglC and tul4 that are specific for F. tularensis as described in Kugeler et al. (2005) and Versage et al. (2003). All reactions were performed in a final volume of 20 μl and contained LightCycler FastStart DNA Master HybProbe mix (Roche, Mannheim, Germany) at a 1× final concentration, 500 nM forward and reverse primers, 100 nM probes, and 1.25 U of uracil-DNA glycosylase per reaction. For the iglC and tul4 the final MgCl2 concentration was 4 mM, and for the ISFtu2 assay, the final concentration was 5 mM. Real-time PCR cycling conditions were: 50 °C for 2 min; 95 °C for 10 min; 45 cycles of: 95 °C for 10 s, 60 °C for 30 s; and 45 °C for 5 min.

PCR amplification of the cytochrome b gene was used to query the DNA from the ticks for determining the hosts of their prior blood meals using the primers UNFOR403 and UNREV1025 (Kent & Norris, 2005; Lah et al., 2015). PCR reactions were conducted using 2X Taq PCR Master Mix (Qiagen, Valencia, CA) with primer concentrations at 0.2 μM, 8 μL of template per reaction and a total reaction volume of 40 μL. PCR cycling conditions were: denaturation at 94 °C for 3 min followed by 35 cycles of 94 °C for 1 min, 52 °C for 1 min, and 72 °C for 1 min; then final extension at 72 °C for 7 min before holding the PCR products at 4 °C.

For the bacterial community analysis, a segment of the conserved bacterial 16S rRNA gene was amplified from the individual tick DNA extractions using universal primers 515F and 806R that flank the V4 region (Caporaso et al., 2012). The 806R primers also contained a unique 12-nucleotide Golay “barcode” for each sample that allowed us to pool the PCR products from all the samples into one Illumina MiSeq sequencing run but then to identify sequences derived from each individual tick. PCR reactions were conducted in a total volume of 40 μL using Taq98® Hot Start 2X Master Mix (Lucigen, Middleton, WI, USA) with primer concentrations at 0.2 μM. PCR cycling conditions were: denaturation at 98 °C for 2 min followed by 35 cycles of 98 °C for 30 s, 55 °C for 30 s, and 72 °C for 1 min; then final extension at 72 °C for 10 min before holding the PCR products at 4 °C. The PCR products were visualized under UV light on 1% agarose gels stained with ethidium bromide before being normalized and sequenced on an Illumina MiSeq instrument by The Scripps Research Institute DNA Array Core Facility using their standard protocols (TSRI, San Diego, CA, USA).

Computational and statistical analyses

The sequence data was analyzed using the QIIME (Quantitative Insights Into Microbial Ecology) version 1.8.0 software program (Caporaso et al., 2010b). Raw sequence data was demultiplexed into samples by barcode and filtered by mean quality score below 25, homopolymers greater than 6, uncorrected barcodes, barcodes not found in the mapping file, chimeric sequences and mismatched primers. Sequences were grouped into operational taxonomic units (OTUs) at the 97% sequence similarity level using UCLUST (Edgar, 2010) and a consensus taxonomic classification was assigned to each representative OTU using the UCLUST classifier with a Greengenes 13_8 reference database (DeSantis et al., 2006) in which at least 90% of the sequences within the OTU matched the consensus taxonomic classification 16S rRNA gene. Sequences were aligned using PyNAST (Caporaso et al., 2010a) against the Greengenes 13_8 reference core set and a phylogenetic tree of the OTUs inferred using FastTree (Price, Dehal & Arkin, 2010). In order to remove spurious OTU’s and samples with low numbers of sequences, OTU’s that occurred only once in the data and samples with less than 150 OTUs were removed. Rickettsia, Francisella and other selected taxonomic sequence identifications were crosschecked against the NCBI nucleotide database using BLASTn. Sequence, OTU table and map files can be downloaded from Figshare: 10.6084/m9.figshare.2056275, 10.6084/m9.figshare.2068644, and 10.6084/m9.figshare.2056272, respectively.

The OTU dataset was rarefied to an even sampling depth of 150 and weighted and unweighted UniFrac distance measures between all pairs of microbial communities were calculated and visualized by principal coordinate analyses (PCoA) (Lozupone & Knight, 2005). Rarefying at 1,500 even sampling depth resulted in similar results. Several analyses were performed to determine possible factors related to microbiome differences observed within the ticks and if interference between bacteria was observed. To determine if microbial profiles were consistent with the hypothesis of interference between bacteria, the Pearson product-moment correlation coefficient (PPMC) was calculated using R to determine if a statistically significant relationship existed between the number of sequences of Rickettsia and Francisella found in the various locations. Faith’s phylogenetic diversity measure (PD) was used to compare the alpha diversity between male and female ticks. Unlike other ecological diversity metrics (Shannon, chao1) that rely on species counts, Faith’s PD is calculated based on phylogenetic tree branch lengths which captures an additional aspect of diversity, namely evolutionary diversity. To elucidate which bacteria in the microbiome were associated with the presence of SFGR (as identified by rompA and IGR sequences), Random Forest supervised learning was performed in QIIME using 1,000 trees and 10 times cross validation. The ratio of Rickettsia to Francisella was classified as high (ratio > 2), even (0.5–2) or low (<0.5) in each tick. Correlations between non-Rickettsia, non-Francisella genera in the tick microbiomes and the Rickettsia to Francisella ratios (high, even, or low) was determined via a Kruskal–Wallis H test. Associations between microbiome phylogenetic distances with physical separation of the sampling locations was investigated by comparing microbiome UniFrac measures to the physical distances between sampling locations using the Isolation by Distance (IBD) web service http://ibdws.sdsu.edu/ĩbdws/distances.html (Jensen, Bohonak & Kelley, 2005). IBD tests the linear relationship between geographic distance and genetic distance of a population or, in our case, geographic distance and the microbial community phylogenetic distance. It uses a pairwise Mantel test to assess the relationship between location and UniFrac phylogenetic distances. To determine which of the abundant genera were responsible for differences in UniFrac measures between locations, OTUs that occurred in less than 10% of the samples were removed and the null hypothesis that abundances of OTUs were the same for all locations was tested using a Kruskal–Wallis H test in QIIME. A Procrustes least squares orthogonal mapping analysis was performed in QIIME to determine if the beta diversity of Rickettsia and Francisella populations was similar to non-Rickettsia non-Francisella populations with respect to location (Gower, 1975). Procrustes analysis is a statistical scaling method that transforms multidimensional shape data, in this case, beta diversity matrices, into maximal superimposition (least squared distances) to determine the concordance between the matrices. Furthermore, Analysis of Similarity (ANOSIM), which compares the ranked Bray–Curtis similarity between and within groups, was used to determine whether microbial population beta diversity between locations differed significantly. We also compared inter-and intra-subject microbial community variability using PERMANOVA (Anderson, 2001), a non-parametric multivariate analysis of variance test that employs a permutation procedure to test the null hypothesis that there is no difference between and within subjects. We used Bray–Curtis distance and 999 permutations in R studio (Version 0.99.893, vegan package).

SourceTracker was used to compare the tick microbial profiles to microbiome datasets of dog, fish, iguana, human, pigeon, rat, and soil. SourceTracker is a tool that uses Bayesian methods to predict the source(s) of microbial communities in a set of samples (sink) (Knights et al., 2011). (The inclusion of human microbiome information, especially skin, also allowed us to test for investigator-introduced contamination since skin bacteria is the most common source of laboratory and indoor contamination.) To test for sources of the tick microbiomes (sink), microbial source tracking was performed on the merged sink and source OTU file. A logistic regression was perform using the general linear model function in R, was used to determine if Rickettsia presence or absence (binary outcome) in the tick was associated with a particular host source. SourceTracker was also used as a quality control measure to identify possible sample contamination. SourceTracker version 1.0 was implemented in QIIME (version 1.9.1) with default settings. As source datasets, we used publicly available sequence data in QIITA (https://qiita.ucsd.edu/) that included 16S rRNA data from a wide range of samples such as canine skin, mouth, and feces (Study ID 1684), human skin, mouth and stool (Study ID 1684), soil (Study ID 1684, 10363), fish, frog, iguana, pigeon, and rat skin (Study ID 1748) and negative water controls (Study ID 10363) as sources. All source and sink samples were sequenced using Illumina and the same 16S rRNA V4 region primers.

Results

Four hundred seventy-four adult D. occidentalis ticks were collected. No immature ticks were caught. Although no ticks were positive for Francisella tularensis, 39 ticks (8.2%) were positive for R. rhipicephali and 12 (2.3%) were positive for R. philipii 364D as identified by sequencing of the rompA gene and IGR. No significant difference in infection rate between male and female ticks by R. rhipicephali and R. philipii was observed (Fisher’s exact test; P = 0.47). From this group, 114 ticks were selected for Illumina sequencing. Amplification and gel electrophoresis of the V4 segment of the 16S rRNA gene produced visible PCR products of the expected 300 bp size from all ticks, while negative PCR and DNA extraction controls yielded no visible bands and were not sequenced. After quality filtering, 102 ticks remained: 44 positive for SFGR (as identified by rompA and IGR sequencing) and 59 negative for SFGR (forty-five male and fifty-seven female) from the four locations (Table 1); the total number of sequences was 6,799,927 with sample depths ranging from 2013 to 250403 reads (Table S1). Clustering sequences at the 97% level of similarity and discarding OTUs that occurred only once yielded 105,174 different OTUs and 535 different taxa including one unassigned taxon. Rickettsia and Francisella genera were the most prevalent genera present in the ticks, representing 46.8% and 41.4% of all genera, respectively. The next most frequently occurring genera were Sphingomonas (3%), Methylobacterium (1%) and Hymenobacter (0.4%) (Fig. 1).

Table 1 Tick collection locations, number of ticks infected with Spotted Fever group Rickettsia, and number of male and female D. occidentalis ticks collected at each location.

Location	GPS Coordinates	R. rhipicephali	R. philipii	Negative	M/Fa	
Escondido Creek	33.060700, −117.179500	7	1	9	8/9	
Lopez Canyon	32.906776, −117.202964	14	9	22	23/22	
Mission Trails	32.834444, −117.045833	4	1	19	7/17	
Peñasquitos Canyon	32.938638, −117.130351	7	1	8	7/9	
Notes.

a No statistically significant association between SFGR infection and male versus female, Fisher’s exact test; P > 0.5. M, male; F, female.

One Rickettsia sp. (OTU 83718) accounted for 89% of all Rickettsia OTUs and matched 99–100% to R. rhipicephali (GenBank accession numbers CP013133.1, NR_074473.1, CP003342.1, NR_025921.1, and U11019.1). It also had 99% identity to other Rickettsia; however, E values were 100× higher to these other Rickettsia sequences. The second most abundant OTU (553807) accounted for 0.7% of all Rickettsia OTUs and was 99% identical to several different R. rickettsii strains including R. philipii str. 364D (GenBank: NR_074470.1) as well as other strains of R. rickettsii (including GenBank accession numbers CP006010.1, NR_102941.1, and CP003311.1). All other Rickettsia OTUs comprised less than 0.09% of total Rickettsia OTUs. OTU 840032 comprised 87.4% of all Francisella OTUs and matched 100% with Francisella-like endosymbiont (FLE) of D. occidentalis (GenBank accession numbers AY805304.1, and AY375402.1). The next closest matches were Francisella endosymbionts of other tick species D. albipictus, D. andersoni and D. variabilis (GenBank accession numbers GU968868.1, FJ468434.1, and AY805307.1, respectively). The next most abundant Francisella OTU (399541) (GenBank acc. KU355875.1, this paper) accounted for 3.1% of all Francisella OTUs and matched 97% with gene sequences of endosymbionts previously determined from a spectrum of Dermacentor species including Dermacentor occidentalis (AY375403.1), D. albipictus (GU968868.1), D. variabilis (AY805307.1), D. nitens (AY375401.1) and D. andersoni (AY375398.1). All other Francisella OTUs accounted for less than 0.4% of the total Francisella OTUs.

Figure 1 Most abundant bacterial genera detected in D. occidentalis from four different locations in San Diego County.

Female ticks had significantly less microbial diversity (alpha diversity) than male ticks as measured by Faith’s Phylogenetic Diversity which measures diversity based on phylogenetic tree lengths (Faith’s PD, two sample t-test; t = 3.63, P < 0.01; Fig. 2). Although there was no significant difference between the mean number of Rickettsia and Francisella sequences in male versus female ticks (Student’s t test P = 0.36, 0.06, respectively), Rickettsia and Francisella endosymbionts comprised a greater percentage of the microbiome of female ticks than male ticks 74.9% and 60.1%, respectively (Student’s t test P = 0.02; Fig. 3).

Figure 2 Boxplot of microbiome alpha diversity in D. occidentalis ticks measured by Faith’s phylogenetic diversity (PD) whole tree as implemented in QIIME of male and female D. occidentalis.

Faith’s PD, two sample t-test, male versus female; t = 3.63, P < 0.01.

Figure 3 Percent abundance of taxa that comprise at least 1% or greater of the total microbiome in male and female ticks.

Escondido Canyon had lower average alpha diversity than Lopez and Peñasquitos canyons, P =0.05 (Fig. 4). Beta diversities of unweighted and weighted tick microbiomes had small but statistically significant associations with location as measured by analysis of similarity (ANOSIM) of UniFrac distances and visualized on Principal coordinate analysis (PCoA) (Figs. 5A and 5B). When only Rickettsia and Francisella were assessed for association with location, ANOSIM results were not statistically significant (ANOSIM, unweighted UniFrac; R =  − 0.06, P = 0.92; ANOSIM, weighted UniFrac R = 0.02, P = 0.13). However, in order to determine if location influenced the non-dominant species separately from the dominant Rickettsia and Francisella endosymbionts, Rickettsia and Francisella were removed from the data and the analysis was repeated. After Rickettsia and Francisella were removed, the remaining microbiome association with location was low but statistically significant (ANOSIM, unweighted UniFrac; R = 0.20, P < 0.01; ANOSIM, weighted UniFrac; R = 0.28, P < 0.01). Procrustes analysis also demonstrated that the beta diversity of microbiomes in which Rickettsia and Francisella were removed had a different association with location than Rickettsia and Francisella endosymbionts (error, M2 = 0.91, P < 0.01). Isolation by distance (IBD) analysis using unweighted UniFrac distances that incorporated all members of the microbiome revealed little geographic IBD (Mantel test, unweighted UniFrac, R = 0.09, P < 0.01). However, a pattern of IBD was significant after excluding Rickettsia and Francisella (Mantel test, unweighted UniFrac, R = 0.14, P < 0.01). (Unlike weighted UniFrac, unweighted UniFrac distances only incorporate the presence or absence of microbial taxa and to not take into account the abundance of the particular taxa. This allowed us to focus on the taxonomic differences among locations rather than the abundance differences of particular taxa. In a separate analysis, we found significant differences in the relative abundances of three bacterial genera, Nevskia, Curtobacterium and Sphingomonas, between locations (Kruskal–Wallis H = 25.7, 24.2, 22.9; Bonferroni corrected P < 0.01, respectively). Ticks in Peñasquitos and Lopez Canyons had higher abundances of Nevskia than ticks collected in Mission Trails. Peñasquitos and Lopez Canyons ticks had higher relative abundances of Curtobacterium and Sphingomonas than ticks in Escondido Creek and Mission Trails (Table 2).

Figure 4 Boxplot of microbiome alpha diversity in D. occidentalis ticks measured by Faith’s phylogenetic diversity (PD) whole tree as implemented in QIIME of four different hiking areas in San Diego County.

Stars indicate statistically significant differences between samples; Faith’s PD, two sample t-test, Escondido Creek versus Lopez Canyon; t =  − 3.28, P = 0.02; Escondido Creek versus PQ, t =  − 3.31; P = 0.04; other comparisons were not statistically significant. PQ = Peñasquitos Canyon.

Figure 5 (A) Unweighted beta diversity of D. occidentalis microbiomes at four different locations in San Diego County. ANOSIM, unweighted UniFrac; R = 0.14, P < 0.01. (B) Weighted beta diversity of D. occidentalis microbiomes at four different locations in San Diego County. ANOSIM, weighted UniFrac; R = 0.12, P = 0.01.

Table 2 OTUs and genera associated with different locations.

OTU	Genus	Ha	Pb	Escondido Creekc	Mission Trailsc	Peñasquitos Cync	Lopez Cync	
73481	Nevskia	25.7	0.0002	1.59	0.04	2.31	1.09	
643513	Curtobacterium	24.2	0.0004	0.18	0.25	3.06	1.91	
489455	Sphingomonas	22.9	0.0007	0.12	0.13	3.69	2.09	
Notes.

a Kruskal–Wallis H value.

b Bonferroni correction (Bonferroni correction is used to reduce the chances of obtaining false-positive results (type I errors) when multiple pair wise tests are performed on a single set of data because the probability of identifying at least one significant result due to chance increases as more hypotheses are tested).

c Average number of OTU occurrences per sample.

Rickettsia and Francisella were negatively correlated in the ticks (Pearson’s product moment correlation; R =  − 0.44, P < 0.01; Fig. 6). In order to assess whether the tick microbiomes were predictive of infection with spotted fever group Rickettsia (as determined by real-time PCR of the rompA gene and IGR sequences), a Random Forests supervised learning analysis using 1,000 trees and 10× cross validation was performed on the OTU dataset minus Rickettsiaceae and Rickettsia OTUs. The ratio of baseline error to the estimated generalization error was 8.8 (i.e., more than 8 times greater than random chance). The most predictive OTU was the FLE OTU 840032 and it accounted for 13% of the model. OTUs 866436 and 639277 each accounted for 3% of the model and the closest database matches to it were the Firmicutes Geobacillus and Aeribacillus (Geobacillus), respectively (Minana-Galbis et al., 2010). Non-Rickettsia, non-Francisella bacteria associated with Francisella to Rickettsia > 2 (range 2.4–119.0) were Planococcaceae and Geobacillus (Kruskal–Wallis test; H = 23.8, 14.2, Bonferroni P < 0.001 and P = 0.011, respectively). In addition, PERMANOVA with Bray-Curtis distances was used to test for the impact of tick sex, collection site and endosymbiotic infection status on the composition of microbial communities between samples (Anderson, 2001). These environmental differences showed a significant influence on the tick associated microbial communities. SFG (P = 0.001), tick sex (P = 0.021), location (P = 0.001), and Rickettsia to Francisella ratio (P = 0.008) all had significant effects on the bacterial communities.

Figure 6 Rickettsia and Francisella OTU abundance in D. occidentalis ticks in San Diego County.

Plus signs (+) indicate ticks infected with R. rhipicephali and slashes (∕) indicate ticks infected with R. philipii 364D. Pearson product moment correlation; R =  − 0.44, P < 0.01. (A) Mission Trails; (B) Lopez Canyon; (C) Escondido Canyon; (D) Peñasquitos Canyon.

Amplification of vertebrate cytochrome b gene was attempted to determine the origin of the ticks’ host blood meals; however, no cytochrome b was amplified from the ticks. This may have been due to ticks being captured before feeding as they were questing for a blood meal. SourceTracker analysis revealed that 31% of ticks had microbiomes that were between 1.1 and 27.4% similar to dog skin microbiomes (Table S2). Ticks negative for R. philipii or R. rhipicephali were more likely to have microbiomes similar to dog skin than ticks that were infected with R. philipii or R. rhipicephali (Generalized Linear Model; P = 0.023). Sphingomonadaceae, Oxalobacteraceae, and Comamonadaceae were the most abundant families of bacteria shared between tick and dog skin microbiomes. The tick microbiomes were less than 1% similar to microbiomes of the skins of fish, iguana, pigeon, rat, and human, as well as human oral, plant and soil microbiomes (Table S2).

Discussion

D. occidentalis is one of the most common tick species found in San Diego and is a vector of human pathogens including Francisella tularensis and Rickettsia philipii. This is the first study of the microbiome of D. occidentalis ticks using NGS technologies to examine pathogen interference within the tick microbiome. Although Francisella tularensis has been detected previously in ticks in San Diego (Kugeler et al., 2005), none of the ticks harbored this bacterium or genera of other recognized zoonotic tick-borne pathogens such as Borrelia, Anaplasma, Ehrlichia, Babesia or Bartonella; however, a low percentage of the ticks were infected with spotted fever group Rickettsia: 2.5% with R. philipii and 8.2% with R. rhipicephali. This is a slightly lower prevalence of R. philipii than surveys of ticks performed in Orange, Riverside, Los Angeles, Santa Barbara and Ventura counties north of San Diego, that reported an overall 7.5% prevalence of R. philipii (Wikswo et al., 2008) but is within the range of R. philipii prevalence reported from northern California of 0.4–5.1% (Lane et al., 1981; Philip, Lane & Casper, 1981). Similar to other tick species, the microbiome of D. occidentalis was dominated by Proteobacteria, primarily Rickettsia or Francisella, with lesser amounts of Sphingomonas, Methylobacterium and Hymenobacter (Bacteroidetes). These last three genera are all decomposer microbes found in the soil and except for Hymenobacter, have been detected in other tick microbiome studies. Even though the ticks were washed multiple times before DNA extraction, the possibility that some of these represent surface bacteria cannot be completely excluded. Although not performed in this study, removal of any OTUs detected by sequencing a sterile water negative control would also improve the sequence quality of future analyses. However, it is worth noting that members of the genus Sphingomonas were also found in two different studies of Ixodes tickes (Van Treuren et al., 2015), including one that studied larvae, suggesting that this genus may be arthropod-associated.

Although 58 of the ticks were negative for SFGR by real-time PCR of the rompA gene and IGR, all of the ticks contained OTUs whose partial 16S rRNA gene segments aligned with SFGR in GenBank. The cause of this discrepancy may be due to the increased sensitivity of the Illumina sequencing platform compared to real-time PCR of rompA and IGR sequences and/or the presence of other Rickettsia spp. with highly conserved 16S rRNA genes but that lack rompA and IGR sequences complementary to the PCR primers used. Analysis of other genes would be required to resolve them at the species level (Eremeeva, Yu & Raoult, 1994; Regnery, Spruil & Plikaytis, 1991). Additional data support that more than two different Rickettsia species were present within the tick population tested. R. rhipicephali was detected by real-time PCR of the rompA gene and/or IGR in ticks that had OTU 837189 counts greater than 5900/tick, except for two ticks, T14-0667 and T14-0769 that had high OTU 837189 counts of 73,527 and 53,714, respectively, but were negative for R. rhipicephali. Similarly, R. philipii was detected in ticks with OTU 553807 counts ranging from 11 to 2,158, except for one sample, T14-0667, that had 884 counts of OTU 553807 yet was negative for R. philipii by real-time PCR of rompA gene and IGR. These findings are consistent with the presence of species of Rickettsia different from R. rhipicephali and R. philipii that could not be discriminated by the partial 16S rRNA gene or rompA and IGR sequences. The two most abundant Francisella OTUs, 840032 and 399541, accounted for over 90% of all Francisella OTUs and were 100% identical to Francisella-like endosymbionts (FLE) of D. occidentalis (GenBank accession numbers AY805304 and AY375402 for OTU 840032, and KU355875 for OTU 399541). Taken as a whole, these results are consistent with tick co-infection with a mixture of Rickettsias and FLEs.

The number of unique OTUs detected in D. variabilis was 6.4 times higher than found in a study of Ixodes ticks, although, sequence depth was approximately two times greater in our study and, as noted, the vast majority of the OTUs occurred at very low frequencies (Van Treuren et al., 2015). OTUs that occurred only once in the data were removed. However, a presence threshold (i.e., requiring OTUs to be present in more than one tick) was not applied so that rare species that contributed to microbiome differences between locations would not be filtered out. This resulted in a large number of rare OTUs analyzed in our data, although, the interesting inverse relationship of the dominant endosymbiont species was not affected.

The low frequency OTUs found in the study could have partially been the result of contamination.Although negative water controls did not yield visible products when subjected to 16S rRNA PCR, these controls were not subjected to NGS sequencing. SourceTracker analysis did not find evidence of skin contaminants commonly found in reagents or samples (Hewitt et al., 2013). SourceTracker was originally designed to analyze laboratory and built environment contamination and skin bacteria are the most common contaminants in these environments (Kelley & Gilbert, 2013), suggesting that investigator introduced contamination was not a major contributor to the low frequency OTU numbers. Nonetheless, in future studies would be much improved by the incorporating negative water extraction controls carried though to NGS. Future studies would also be significantly improved by following a more rigorous contamination protocol procedure such as that outlined by Kozich et al. (2013): https://github.com/SchlossLab/MiSeq_WetLab_SOP/blob/master/MiSeq_WetLab_SOP_v4.md.

Similar to Ixodes scapularis and Amblyomma americanum ticks, female D. occidentalis ticks harbored a less diverse array of bacteria than males (Fig. 2) (Ponnusamy et al., 2014; Van Treuren et al., 2015). Endosymbionts belonging to Rickettsia, Coxiella, Francisella and Arsenophous genera have been found in different tick species and are thought to interfere with and partially exclude other bacteria and pathogenic forms of closely related organisms from transovarial transmission leading to lower alpha diversity in female ticks than males (Burgdorfer & Brinton, 1975; Macaluso et al., 2002; Niebylski, Peacock & Schwan, 1999; Noda, Munderloh & Kurtti, 1997; Reinhardt, Aeschlimann & Hecker, 1972; Telford III, 2009). In D. occidentalis, a higher percentage of Rickettsia and Francisella in the microbiomes of female ticks than male ticks may have similarly decreased species richness in female ticks compared to males (Fig. 2).

The beta diversity of the endosymbionts and non-endosymbionts differed with respect to location. Although non-endosymbionts demonstrated a small association with location, geographic association was not observed by the Rickettsia and Francisella endosymbionts. In addition, Procrustes analysis results demonstrated that Rickettsia and FLE beta diversities had different relationships to geographical locations than the other microbiome components, illustrating that different factors shape Rickettsia and FLE components of the D. occidentalis microbiome than other non-endosymbiont microbiome members. One factor that appeared to contribute to the geographical differences in the non-endosymbiont microbiome was isolation by distance. Geographical differences in bacterial community composition in the same hematophagous insect species has been seen in fleas and ticks, however, the causes are not completely known (Jones, Knight & Martin, 2010; Van Treuren et al., 2015). Differential geographic localization of Nevskia, Curtobacterium and Sphingomonas, genera that are associated with environmental sources such as the air-water interface (Nevskia) and soils (Curtobacterium and Sphingomonas), may be the result of differences in soil microbial ecology at each location (Van Treuren et al., 2015). Alternatively, non-endosymbiont microbiome differences could be the result of stochastic or different populations of ticks at each location. In contrast, the dependency of Rickettsia and Francisella endosymbionts on their D. occidentalis host may have restricted the degree of variation that population separation could impart upon these endosymbionts (Budachetri et al., 2014).

One of the primary hypotheses of this study was to determine if negative associations between bacteria, suggestive of interference, occurred within ticks especially with respect to pathogens. Indeed, a strong inverse relationship was observed between Rickettsia and FLE infection (Fig. 6) and a Random Forests supervised learning model successfully predicted the absence of SFGR within the ticks. Not surprisingly, FLE OTU 840032 contributed most to the model. FLE and different uncategorized Rickettsia co-infection in ticks has been previously observed but not enumerated (Niebylski et al., 1997; Scoles, 2004) and partial interference between co-infection by different Rickettsia species has been demonstrated (Burgdorfer, Hayes & Mavros, 1981; Macaluso et al., 2002). Although the quantitative 16S rRNA gene sequence data of FLE and Rickettsia co-infection in this study do not directly measure interference between the organisms, they are consistent with interference between FLE and Rickettsias and require further experimental studies for confirmation.

The mechanisms by which Rickettsia and Francisella interfere with each other in co-infections are not known and there can be many kinds of interaction, direct or indirect, between microbes and the tick host which can be influenced by the infection behavior of the different bacterial species. Although the localization of R. rhipicephali and R. philipii within ticks has not been determined, FLEs have been found in female tick reproductive tissues and hemolymph (Goethert & Telford, 2005; Scoles, 2004). In addition, non-Francisella bacteria were also associated with low Rickettsia to Francisella ratios. Planococcaceae and Geobacillus were associated with greater abundance of Francisella relative to Rickettsia within the ticks. Although blood meals of the ticks could not be detected by amplification of vertebrate cytochrome b gene from the ticks, 31% of the tick microbiomes had microbiome components similar to canine skin which may suggest the source of a prior blood meal if they incorporated some of the skin flora into their own microbiome as has been shown with host blood microbiomes following feeding (Zhang et al., 2014). Use of SourceTracker for comparison of tick and skin microbiomes is a novel approach and, interestingly, generalized linear models showed that ticks with canine skin microbiome components were less likely to be infected with Ricketssia which is consistent with R. rhipicephali and R. philipii being endosymbionts without a canine host.

Conclusions

The results of this study suggest that FLE and Rickettsia endosymbionts partially exclude each other in co-infections of the same D. occidentalis tick. Although interference between Rickettsia co-infections has been known for many years, this is the first study that suggests possible exclusion between different endosymbiont genera in ticks. Whether FLEs can be shown to inhibit Rickettsia co-infection in the laboratory, the mechanisms for their interaction and whether they could be propagated through a tick population as a means to render ticks unable to vector pathogenic Rickettsia are intriguing prospects that warrant further investigation. In addition to more mechanistic investigations, future research should also apply approaches such as shotgun metagenomics that can be used to assemble long sequence reads and even complete genomes from whole microbiome samples for more accurate species and strain identification.

Supplemental Information

Table S1 Spotted Fever group Rickettsia and number of 16S rRNA gene sequences from each tick

Spotted Fever group Rickettsia identified by rompA and IGR sequencing and total number of individual 16S rRNA gene sequences from each tick.

Click here for additional data file.

Table S2 SourceTracker results for D. occidentalis microbiomes from San Diego County

EC, Escondido Creek; LC, Lopez Canyon; MT, Mission Trails, PC, Peñasquitos Canyon.

Click here for additional data file.

Additional Information and Declarations

Competing Interests

Author Contributions

DNA Deposition

Data Availability

The authors declare there are no competing interests.

Nikos Gurfield conceived and designed the experiments, performed the experiments, analyzed the data, contributed reagents/materials/analysis tools, wrote the paper, prepared figures and/or tables, reviewed drafts of the paper.

Saran Grewal and Lynnie S. Cua performed the experiments, analyzed the data, contributed reagents/materials/analysis tools, wrote the paper, reviewed drafts of the paper.

Pedro J. Torres performed the experiments, analyzed the data, contributed reagents/materials/analysis tools, wrote the paper, prepared figures and/or tables, reviewed drafts of the paper.

Scott T. Kelley conceived and designed the experiments, analyzed the data, contributed reagents/materials/analysis tools, wrote the paper, prepared figures and/or tables, reviewed drafts of the paper.

The following information was supplied regarding the deposition of DNA sequences:

GenBank: KU355875.1.

The following information was supplied regarding data availability:

Gurfield, Nikos; Kelley, Scott (2015): Dermacentor occidentalis microbiome 16S rRNA gene sequences from San Diego County. figshare.

https://doi.org/10.6084/m9.figshare.2056275.v2;

Gurfield, Nikos; Kelley, Scott (2016): Dermacentor occidentalis 16S rRNA OTU table. figshare.

https://doi.org/10.6084/m9.figshare.2068644.v1;

Gurfield, Nikos (2015): Mapping file for Dermacentor occidentalis 16S rRNA microbiome dataset from San Diego County. figshare.

https://doi.org/10.6084/m9.figshare.2056272.v2.

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
