# Peer review of "Endosymbiont interference and microbial diversity of the Pacific coast tick, Dermacentor occidentalis, in San Diego County, California"

_PeerJ, doi:10.7717/peerj.3202_

## Round 0.1 · original submission · Major Revisions

Each reviewer thought the study question was relevant and interesting but as you will see from the reviews, each reviewer had their own, distinct criticism of the manuscript. I agree that any methods - including statistical analyses - need to be better explained and that the limitations of the approach (16S rRNA gene amplicon studies) must be better incorporated into the interpretation of these data. However, I realize that there is a great quantity of suggested edits and alternative analyses provided here by these expert reviewers. If you choose to not implement these changes, make sure to make the case in your response to reviewers.

·

Basic reporting

The manuscript provides valuable information about bacteria communities in the Pacific coast tick. There is value in which molecular assays can or cannot provide information about the bacterial communities associated with ticks. The methodology used is current with microbiome characterization, but clarifications in the methodology are needed. The results need to focus on the data generated, with the limitations of the methods used. Finally, the discussion should provide a more critical and transparent evaluation of their data in the context of other tick microbiomes.

The conclusion section was out of place. While the idea of using symbionts to control the spread of pathogens is intriguing and being developed in the Wolbachia system, the authors did not prepare the reader for these concluding remarks. A better conclusion would focus on the findings and limitations of the study given the data the authors actually present.

Experimental design

It is not clear how exactly the authors used the preliminary PCR queries for Rickettsia and Francisella bacteria to inform their Illumina sequencing approach. This should be clarified—it is unclear if ticks were pooled per site, or if individual ticks were sequenced. It is surprising that such low frequencies of Rickettsia and Francisella infection were found by PCR, but such a large percentage of the OTUs were close in identity to Rickettsia and Francisella. It is also unclear what percentage of individual ticks was infected with Rickettsia or Francisella, as revealed by Illumina sequencing. The authors should clarify their methods to address this issue.

Negative controls were run in parallel to the DNA extractions (Line 122). It is unclear if these were sequenced in parallel with the tick samples and how the negative controls were used in the data analyses. Many of the differences in site (Sphingomonas, Methylobacterium) may be common contaminants. As the authors address (Lines 392-397) they may just be remnants of the soil community, and while care was taken to surface sterilize the ticks, these potential soil microbes may not be biologically relevant to the tick. The authors should better discuss these bacteria in the context of the biology of the tick to show how the bacteria actually contribute to differentiation by site, and not just an artifact of contamination in the lab. It would also benefit their argument if they can provide evidence for isolation by distance in bacterial communities from other studies.

Validity of the findings

The authors stated that Rickettsia and Francisella might interfere with each other (Line 67-76). Understanding competition between endosymbionts is an open question in the field of host-associated microbial ecology. However, 16S rRNA profiling provides relative abundances of sequences, but does not provide measures of competition or an accurate estimate of competitive interference. RNA expression would address this question by looking at transcriptionally active bacteria, and the authors should clarify their conclusions to address this limitation of 16S rRNA profiling (lines 402-413).

SourceTracker was a nice idea to figure out what contributed to the bacterial communities in the ticks. However, again, the authors discuss just the co-occurrence of the bacteria (lines 428-437), but the bacteria also seem to be biologically irrelevant to both ticks and dogs. Limitations of this approach should be discussed. Primarily, other studies have sequenced feeding ticks and the blood of the host the ticks was found to determine sources of the tick microbial community. The current discussion of the results from SourceTracker does not seem biologically relevant. I recommend the authors compare their results to those matched with tick and host blood as a better way to examine the source of host-associated microbiomes.

Additional comments

They authors do a nice job of motivating the importance to public health of microbial surveys in ticks, and provide a survey of the both the pathogens and potential symbionts commonly found in ticks. There is a lot of value in the negative results presented. It is important to the community that identifying the Rickettsia/Francisella symbionts and identification of digested blood meal is difficult with standard PCR.

The authors find that male ticks have more microbial diversity than female ticks, and this is common to other tick microbiome studies. The authors should clarify the life stage of ticks (Line 102). This result is interesting, although the reason is unknown. It is an important consistency in the tick microbiome literature where inconsistency and high variation is common.

Reviewer 2 ·

Basic reporting

No comments.

Experimental design

No comments.

Validity of the findings

Given the extremely low R-values, I think the effect of geography (especially given that there are only 4 sites) are a bit overstated (see Line 260 and the very low ANOSIM R-value). The figures show the lack of any substantial effect of geography.

Additional comments

I have a few minor comments:

1) Line 19. Replace non-infectious with non-pathogenic

2) Lines 25-26. Might be worth mentioning levels of co-infection of different Rickettsia strains in the same tick here.

3) Line 197. Why such an extremely low rarefaction level (150 sequences per tick)???

4) Line 207-209. What is the 'genetic distance'? UniFrac values? More details on the stats please. Mantel test?

5) Line 242. If the minimum coverage was 2,013, why was the rarefaction (see comment 3) of 150 used?

6) Line 243. That is a huge # of detected OTUs. Usually when I analyze a dataset like this using UCLUST (but with more, diverse samples), I get 10,000 - 30,000 OTUs. Seems VERY high. Sure you did everything correctly?

7) Seems like some relevant literature of other disease vectors (e.g. fleas, mosquitos) is ignored. I guess I can understand if you only want to focus on ticks, but there has been some other work on fleas that seems relevant to this work (see: Jones, Knight, and Martin. 2013. Bacterial communities of disease vectors sampled across time, space, and species; Jones, Bernhardt, Martin, and Gage. 2012. Interactions among symbionts of Oropsylla spp.). Full disclosure: I'm Jones.

Reviewer 3 ·

Basic reporting

This manuscript is in line with the general guidelines for the journal. There are a few small errors in some of the references which can easily be fixed.

Experimental design

The experimental design is good and relates to the authors’ goals. There should be more clearly laid out predictions in relation to the more specific questions they tried to answer, e.g. concerning the impact of location and tick sex on microbial community composition.

Validity of the findings

The findings are good, but many of the statistical methods were not explicitly described which hinders the validity of the claims. It has been pointed out where this needs to be addressed. An additional way to improve the quality of the sequence data used by accounting for sequences in the negative controls is also suggested.

Additional comments

In this manuscript, the authors analyze the bacterial communities of adult Dermacentor occidentalis from four different sites near San Diego, CA. The communities were analyzed in relation to endosymbiont infection, Francisella-like endosymbiont, Spotted Fever Group Rickettsia infection, collection site, tick sex, and relation to possible host microbiome. Their goal was to find evidence of interference competition between the endosymbiont and SFGR pathogen infection. They suggest their results show evidence of the hypothesized competition between these two groups of tick-associated bacteria and found some evidence for variation for bacterial community composition by tick sex and collection site, and perhaps some association with host skin microbiome. The implications are the application of endosymbionts as a tick-borne infection control measure.

The goals of the paper are interesting, and there are some interesting results, but unfortunately the paper seems to be bogged down by many different kinds of statistical analyses that are somewhat hard to understand, or not explained clearly in the methods section, which makes it hard to be convinced by the authors’ arguments. The paper would benefit from more explanation of the bacterial taxa making up the communities found in these ticks and streamlining the analyses in relation to their predictor variables to better highlight their findings. A more direct presentation of the hypotheses and predictions in relation to the statistical approaches would also help guide the reader through the results and better understand their implications. There also seem to be some small errors in some of the references in the text or in the references section (e.g. correct format of author names, italicized Latin names) which should be corrected. For the current state of the manuscript, I would suggest to reject with an invitation to resubmit after addressing reviewer comments.

Specific comments in relation to line numbers are below.

Abstract

Line 20: What kinds of “bacteria” do you mean here? Pathogenic? Endosymbionts? Environmental? Please be more specific.

26-27: How was infection with Francisella quantified? The number of types? Number of sequence copies? For obligate endosymbionts one would assume 100% infection so there could not be sufficient variation to relate to infection with another pathogen. (also see comment on line 298, Figure 4)

31-33: Unless there is an experimental component of the study you cannot say you have found explicit evidence for competitive exclusion. Please change the language to more accurately relate to the association found between these bacteria, not a direction of the interaction.

32: Change position of commas to: “…FLEs and, to a lesser extent, other bacteria interfere with…”

Introduction

40-41: “…three-host tick”? I have not heard this term before, since ticks all need three hosts to transition between each lifestage. A more useful description would be to note it is a hard tick (as opposed to soft tick). Also, change to “vertebrates” or “vertebrate hosts”, since “animals” can include invertebrates.

57: First name of author in reference? Seems to be a mistake.

58-60: Unclear what you mean here, what “Francisella bacteria” do you mean? All Francisella are bacteria.

67-68: Wording of the sentence makes it unclear if you mean interactions between multiple Rickettsia endosymbionts or between them and other sorts of bacteria that infect ticks.

76: A concluding sentence would be useful here. Why does the interaction between endosymbionts and pathogens matter?

89-91: This hypothesis assumes the endosymbionts and pathogens share the same infection site within the tick in order to compete, which is not always the case. Can you give further evidence as to why this should happen in this system? Also, many symbionts are obligate and therefore there would not be much variation in prevalence (infection should be 100%) and would not provide a useful predictor for variation in infection with other microbes. Are you only looking at non-obligate types to address this issue?

95-99: Not sure if you need to present all main results here. What would be useful is an explanation of the broader impacts of this study to describe the motivation behind what you did.

Methods

108: By identifying ticks by sex you imply you only looked at adults. This should be stated along with an explanation as to why this life-stage only was investigated and not also nymphs which could also be infected with pathogens.

124: Should use “SFGR” acronym here?

124-25: Here and in further paragraphs you did not state this was real-time PCR (as was stated in the results). Please make that clear where appropriate throughout the methods section.

127-28: Need to give the primer sequences here? (also further in methods whenever a new primer is listed?) Otherwise please make clear these were taken directly from a previous publication.

153-55: Why were ticks screened for F. tularensis too? Would it not be detected in the next generation methods for some reason? There are also no methods for this given as was for the Rickettsia infection.

157-163: There was no mention of investigating host blood meal in the introduction when presenting the hypothesis so it is unclear why this step is being done.

202: Can you give further explanation of the Procrustes analysis? This is not as common an analysis as the others you describe.

197-216: This section is very list-y and there are a lot of different analyses done. In general, it would be helpful to explain why there are so many types of analyses being done and which relate to what parts of your hypothesis, otherwise it is hard to follow.

Results

232: How many samples were removed by having too few sequences for rarefaction or other quality control reasons? Were sequences from the negative controls removed? How were these NCs accounted for in the analysis?

236: There is no description of the stats done to compare the bacterial communities between tick sexes. This is true of many of the non-QIIME or sequence-based analyses throughout the results section which will be noted below.

242: add “sequences” or “reads” (whichever is most appropriate” to end of this sentence.

244-47: This seems surprising as Fransicella is the obligate symbiont of most Dermacentor and would be expected to make up the highest proportion of sequences. Is your result that Rickettsia was higher surprising, especially given the PCR results showed low prevalence?

249-50: There was no explanation of the Faith’s PD in the methods. Why was this used instead of a more common metric like Shannon Index?

Figure 2: there is no y-axis label on either 2A or 2B.

Figure 4: The stars seem very awkwardly placed. Are you able to use whatever program you used to make the plot insert symbols for significance?

250-51: By “amount” do you mean number of rarefied sequences? It is important to make this distinction since you did not measure absolute amounts of bacteria, but you are using number of sequences as a proxy.

254: Again, there is no explanation of the ANOSIM analysis in the methods (i.e. what predictors were used, what the test stat means, etc).

255: Can you give an explanation of why location is an important factor in the analysis? It is known that SFGR infections are patchy in distribution or based on host community composition, for example? This should also go in the introduction.

258-61: Is there a reason why the rarer taxa in the community would be more useful to analyze between locations?

261-64: Again, could you provide more context or explanation for how to interpret the results of the Procrustes analysis since it is not commonly used?

164-69: There was no description of the Isolation by distance analysis in relation to the Mantel test in the methods. Please add explanation in the methods. Also, why was the unweighted UniFrac tree used here and not weighted, since you described both were constructed.

270-71: Why are these specific genera referenced here in relation to differences in abundance between sites? They were not all listed as the most common genera. Was this analysis done for all genera?

276: This paragraph should be moved to near the beginning of the results to describe the community composition before going into further analyses.

278-82: If there was a 100% consensus sequence match why go into the other next closest matches? Are any of these of significance? Otherwise I would remove this.

298: Were the number of rarefied sequences used in the Pearson’s correlation? (there was no description of this analysis in the methods)

Figure 4: This figure is impossible to read and get any useful information. It is especially difficult because the statistic associated with it was not explained. I would recommend changing to a scatterplot, if the rarefied sequence of Francisella and Rickettsia were used in the analysis, or a boxplot which has infection category (inf/not inf) with one pathogen as the x-axis and rarefied sequence number as the y-axis. Otherwise there is no way to glean a clear message from the figure or relate it to the analysis done on the data.

309: Add a comma between “Non-Rickettsia” and “non-Francisella”.

298-312: Can you pick either the Pearson’s correlation or the Random Forest analysis to test how infection with Francisella is associated with infection with Rickettsia? Are they redundant? It would streamline the results if only one of these was used since they seem to do very similar things.

315: What does “worked” mean in this context?

317: What data is not shown? If ticks were all collected by dragging they should be definition be host seeking.

317-319, Supplemental Table 2: This table is not very useful. What would be more helpful is a figure or table that illustrate what OTUs are shared between the ticks and the host bacterial communities. There is also no Methods explanation of why a t-test was used, with what data, etc.

321-23: Are these genera common to many microbiomes? If they are very common maybe finding them in both host and tick microbiomes isn’t too surprising, but if they are then this is useful information. Information on if any of these genera are also in the negative controls would help answer this question too.

Discussion

346-48: Again, comparison of the sample and negative controls would help address this concern of contamination or importance of environmentally-derived taxa.

350-65: These seem like new results. They should go in the results section. Also, as they are written they are very hard to follow given all the sample and OTU numbers.

How were the OTUs ranked? As currently stated they are hard to understand especially since there is little or no association between the numbers and the taxa in the results. Ranking by OTU abundance may be more intuitive so the numbers of the OTUs that are more often discussed have “easier” numbers (i.e. OTU1, OTU2, etc). Having a supplemental table of the OTUs and their associated consensus sequences should be included for reader reference.

367-71: These also seem like new results and should be moved to the Results section.

373-84: It would be helpful to include a figure similar to Figure 1 comparing the community composition between males and females in addition to the one you currently have on just differences in diversity. It would be expected that females would have a higher proportion of sequences from the endosymbionts due to these infecting the ovaries.

382-84: Can you give any other possible reason why males and females would differ? Would the dominance of endosymbiont in female samples reduce the number of sequences from other taxa, lowering the measure of richness/diversity in females compared to males?

386-400: Compartments? This is the first time this term has been used. This paragraph is overall well written and easy to follow, but it could use some references to support your interpretation of the results.

418-9: Sentence unclear. Francisella and all other microbes discussed here are bacteria, so the use of the word “microbe” seems unnecessarily broad.

419-34: Again, these seem like new results and should go in the Results section. Discussion of what these results mean can remain in the Discussion.

424: Relation to coyote skin seems a lot like speculation and coyotes are not talked about again anytime following, so please remove.

425-27: This sentence needs a reference that there are indeed symbionts that are acquired through dog blood-meals.

436-37: Parallel sequencing of host blood and ticks would be another avenue to pursue this question (i.e. Rynkiewicz et al. 2015 Molecular Ecology, 24:2566-2579)

441-54: Some of the motivations for the study given here should also be presented in the introduction to give context to the study

---

## Round 0.2 · Minor Revisions

As you can see from your reviewers continued concerns, although you addressed many of their initial critiques, the most egregious concern remains and is not adequately addressed. You must address the fact that you did not include a blank or any other control (such as a mock community, for example) in your Illumina run. Contamination is omnipresent - although you may not have observed amplification, this does not mean that your samples are free of 16S rRNA gene contaminants and the blank should have been included in your run (see https://github.com/SchlossLab/MiSeq_WetLab_SOP/blob/master/MiSeq_WetLab_SOP_v4.md for example). Contaminating microbes can come from anywhere (your lab reagents, for example). Additionally, you do not justify your use of all OTUs (including those that are extraordinarily rare and those that are not present across samples). Because you did not set a specific threshold for abundance or remove OTUs not present across a majority or a plurality of your replicate samples, your OTU numbers are likely inflated and because you did not include a negative control, many of these spurious OTUs are likely contaminants. Either make sure to justify this explicitly in the methods or repeat your analysis using abundance and presence thresholds. I suggest you tone down your conclusions based on rare organisms in your dataset.

·

Basic reporting

The authors did a better job of connecting their work to previous work to characterize the tick microbiome. They clarified several aspects of their question, particularly interference between Rickettsia and FLE symbionts. Hypotheses are directly stated in the introduction, which helps guide the reader through their experimental rationale and discussion.

Experimental design

The authors clarified many details in the experimental rationale that were previously vague. They clarified that the microbiome was characterized for individual ticks as well as the life stage sequenced. The statistical analyses were more clearly stated, laying out explicitly the purpose of each test.

I previously noted that the authors do not clarify if they sequence their negative extraction controls—I still believe they should address this point

Validity of the findings

The authors clarified that their methods do not demonstrate competition between the FLE and Rickettsia symbionts, but agree with their definition of interference. They acknowledged that experiments in the lab would be necessary to demonstrate competition between FLE and Rickettsia symbionts. The authors also clarify some of the limitations and novelty of SourceTracker, which strengthens their argument. Finally, the authors better ground their arguments of geographical differentiation of microbiomes in the literature.

The authors provide a more transparent discussion of their findings that contributes to our understanding of how symbionts and microbial communities may contribute to the ability of arthropods to vector diseases.

Additional comments

The authors provide a more transparent discussion of their findings that contributes to our understanding of how symbionts and microbial communities may contribute to the ability of arthropods to vector diseases.

Reviewer 3 ·

Basic reporting

The errors in the previous version have been corrected. I still strongly suggest re-making of one figure (originally Fig 4, Fig 6 in current version) and details are given below.

Experimental design

There is more reference to the associative nature of the results, not causational, which is an improvement. Also, for ease of review, I would request including the updated line numbers of any quoted text in the responses to specific comments. Without line numbers it was much more difficult to follow the authors’ responses.

Validity of the findings

While there has been improvement in clarity of statistical methods (some issues were brought up by multiple reviewers) there are still some issues with the statistics and sequencing analysis that can be improved or addressed further. Specifics are given below.

Additional comments

There have been improvements to the manuscript, however there are still some places that can be improved to improve clarity and analysis rigor.

As was brought up by myself and another reviewer, there seem to be more OTUs than one would expect for this type of analysis and having sequence data from negative controls would have helped eliminate any contamination. In my experience, PCR of 16S rRNA is not sensitive enough to detect with NCs do contain bacterial DNA which can be detected with next-generation methods (sequencing of sterile water NCs revealed many OTUs that were in PCR-negative samples). It is now clear that no sequencing of the NCs was done, so alternately it should be discussed as a way to improve future analyses and an explanation as to why the number of OTUs is so high (lines 423-437).

Multiple analyses done on the data may be able to be replaced with General Linear Models or a PerMANOVA analysis to test the impact of the main variables, tick sex, collection site, endosymbiont infection status, on endosymbiont infection status or microbiome community composition or Diversity. The GLM would be much more parsimonious than using both K-W H tests and t-tests on infection presence/absence or abundance due to the inclusion of multiple variables and being able to adjust error correlation structure to non-normal data (which is likely what you have, since there is likely a skew as to how many samples are infected with each endosymbiont). The PerMANOVA is similar to the ANOSIM analysis, but it can include multiple variables (e.g. tick sex, collection site) and may be able to better describe which variables are important predictors of community composition. These would be straightforward to do with your data in R. Specifics of where these could be done are given below.

Specific comments
97-100: Question concerning infection sites within the tick was not specifically addressed. It would be good to include at least a mention that there can be many kinds of interaction (e.g. direct, indirect) between microbes within the tick and that the function or infection behavior of each bacterial species may determine this.

232-235: This analysis of different endosymbiont ratios on infection (unclear if it is infection at all or sequence abundance with Rickettsia or Francisella) could be one of the variables put into a GLM as opposed to a nonparametric K-W test, as suggested above.

251-53: This is a place where the PerMANOVA could replace/augment the ANOSIM test were variables such as sex and location could be included into one model. Also, as per my previous comment, you should include what predictor variables were included in this analysis, which has not been done).

259-262: Here, % similarity with host source could be included in a GLM of infection or sequence abundance of Rickettsia.

278-81: It is now clear that you did not sequence the NCs, but PCR alone is not sensitive enough to detect low levels of bacterial contamination. It does not need to be brought up here, but in the discussion (lines 423-427) this needs to be brought up as a way to improve sequence quality in future analyses of these sorts of data.

291-311: In your response to my previous question about why additional matches are included if the first is 100% similar you mentioned these matches were included because they were also 100% similar. This was not indicated and should be to validate why you take the time to discuss these matches. There is mention of multiple species/strain infections in the discussion now per this point, which is good.

313-319: There is still no mention of what the Faith’s PD is in the methods section, it first appears in the results section. Your explanation in the response to my previous comment is helpful, something similar/more brief should go in the manuscript text. This is also a place where a GLM could be used to analyze alpha diversity including multiple variables (sex, site) better than a t-test.

327-323: Per my question previously about why analyze just the rare community, I should have worded it as I know the use of looking at the rare community without the dominants to variation in those OTUs that are more likely to vary depending on other variables. However, including a sentence in the MS to this effect would be helpful for others who are less familiar with the tick microbiome.

333-337: While you provided an explanation to me of why just the unweighted UniFrac distance was used, it would be helpful to other readers to have this in the text as well, since both weighted and unweighted have been used in other analyses in the MS.

347-348, Figure 4 (now 6): Sorry, but I have to insist that this figure still does not clearly convey what you would like it to (it is especially not the kind of graph needed to show a correlation). There is a lot of data presented, but not in a way that seems to relate to the relationship you are trying to show, a negative relationship between Rickettsia and Francisella. Here are my detailed suggestions as to how to correct it and still include all the information you have laid out in your 4 bullet points. I hope this is helpful.

- Scatterplot of rarefied Rickettsia sequence against Francesella sequence. Color and/or shape of points would denote which Rickettsia species infected each individual. That there are no points at the origin would denote that each tick is infected with these genera.
- Panel the figure by collection site to show the relationship occurs at each of the four sites.

423-427: Here is where sequencing NCs should be brought in as a weakness in the study but could be used in future analyses to help remove OTUs not useful to the analysis.

461-464, 476-79: Probably don’t need stats results here if they have already been presented in the Results section.

502-504: This language is still a little too strong for correlational data. Suggest replacing “points to” with “suggests possible”.

508-513: I agree with Reviewer 1 who suggested that the mention of biocontrol and Wolbachia seems somewhat unrelated to what has been discussed in the manuscript, since we do not know any mechanism for interaction between these bacteria. The explicit biocontrol discussion and perhaps focus on implications for ecological spread or variation in these endosymbionts.

---

## Round 0.3 · accepted · Accept

You have done an excellent job in responding to the reviewer concerns and I am glad to see the addition of Figure 6 and the inclusion of the analyses suggested (PERMANOVA).